# An Underwater Crack Detection System Combining New Underwater Image-Processing Technology and an Improved YOLOv9 Network

**DOI:** 10.3390/s24185981

**Published:** 2024-09-15

**Authors:** Xinbo Huang, Chenxi Liang, Xinyu Li, Fei Kang

**Affiliations:** 1School of Infrastructure Engineering, Dalian University of Technology, Dalian 116024, China; huangxinbo0722@163.com; 2China Institute of Water Resources and Hydropower Research, Beijing 100048, China; 3China Yangtze Power Co., Ltd., Yichang 443000, China; lixinyu@ctg.com.cn

**Keywords:** underwater crack detection, white balance method, bilateral filtering denoising, YOLOv9-OREPA

## Abstract

Underwater cracks are difficult to detect and observe, posing a major challenge to crack detection. Currently, deep learning-based underwater crack detection methods rely heavily on a large number of crack images that are difficult to collect due to their complex and hazardous underwater environments. This study proposes a new underwater image-processing method that combines a novel white balance method and bilateral filtering denoising method to transform underwater crack images into high-quality above-water images with original crack features. Crack detection is then performed based on an improved YOLOv9-OREPA model. Through experiments, it is found that the new image-processing method proposed in this study significantly improves the evaluation indicators of new images, compared with other methods. The improved YOLOv9-OREPA also exhibits a significantly improved performance. The experimental results demonstrate that the method proposed in this study is a new approach suitable for detecting underwater cracks in dams and achieves the goal of transforming underwater images into above-water images.

## 1. Introduction

Dams, serving as water conservation and hydropower infrastructure, play an important role in flood control, water storage, irrigation, power generation, and ecological environment protection [1,2,3]. Currently, many dams experience structural aging and corrosion during long-term operation, which increases the risk of damage [4,5,6]. Regularly inspecting a dam, comprehensively evaluating and monitoring its safety status, and timely identifying and addressing potential problems are important for the smooth operation of a dam structure.

Damage such as cracks, spalling, erosion, cavities, and wear may occur during the long-term use of dams [7]. These defects significantly affect the safe and stable operation of dams. Among these disasters, cracks have received widespread attention, owing to their proneness, danger, and ability to trigger other disasters [8]. In recent years, deep learning (DL) and computer vision technologies have made significant progress and have led to significant breakthroughs in artificial intelligence and the development of automatic crack detection methods. Devices equipped with high-resolution visual sensors, such as industrial cameras and unmanned aerial vehicles (UAVs), are used to collect numerous images of dam surfaces. The collected images are processed, a deep learning model is trained to learn the damage features in the images, and a crack detection method based on deep learning is then employed [9,10,11,12,13,14,15,16,17,18,19]. However, because of the complex underwater environment, dam-surface crack detection technology cannot be directly applied to underwater crack detection. Underwater images suffer from color distortion, low contrast, a low signal-to-noise ratio, and a lack of detail. Hence, if deep learning technology is applied, it is difficult to obtain and collect datasets of underwater crack areas. An innovative underwater crack detection system is urgently needed to ensure the safe operation of dams.

Currently, research on underwater crack detection is limited. Huang et al. [8] proposed a method to generate underwater dam crack images using the CycleGAN model, a type of adversarial learning network, to convert above-water dam crack images through image-to-image translation. Then, they used the converted underwater dam crack images for dataset annotation and deep learning network training to solve the problem of insufficient datasets. Cao et al. [20] proposed a large-scale underwater crack detection method based on image stitching and segmentation that can adapt to complex underwater environments and perform well in different research fields. Li et al. [21] proposed a comprehensive, pixel-level, underwater structure, multi-defect instance segmentation, and quantization framework based on machine vision and deep learning for hydraulic tunnels. They also developed a video dataset of multiple defects in underwater tunnel lining structures for training and detection. Li et al. [22] proposed a real-time pixel-level automatic segmentation and quantification framework for underwater cracks in dams using a lightweight semantic segmentation network (i.e., LinkNet) and two-stage hybrid transfer learning (TL) for the detection and segmentation of underwater cracks.

Previous studies mostly trained and segmented deep-learning networks based on underwater crack datasets. However, obtaining underwater crack datasets is difficult and time-consuming. Converting underwater crack images into above-water crack images with the same crack characteristics is an effective crack detection method. This method can be used to detect above-water crack images generated after image processing using a more easily accessible water surface crack dataset, which can improve work efficiency. However, new image-processing technology is required to convert underwater images into water surface images.

There are three main differences between the underwater and above-water environments [23]:(1)The optical imaging principle of underwater images is different from that of underwater images. Figure 1a shows that compared with above-water images, the image received by the underwater imaging system is affected by the direct attenuation, forward scattering, and backscattering components [24].(2)Underwater images are likely to be affected by bubbles, sand grains, and other impurities, resulting in more noise than that observed in above-water images [25].(3)The attenuation characteristics of light in water are significantly different from those in air. Figure 1b shows that in an underwater environment, light of different wavelengths has different attenuation rates when propagating underwater [26].

**Figure 1 sensors-24-05981-f001:**
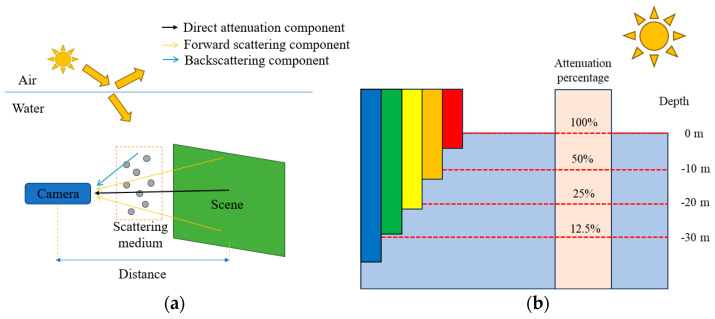
Underwater imaging principle: (**a**) underwater light loss diagram; (**b**) underwater attenuation rate graph of different light rays.

To convert underwater crack images into above-water images, targeted image processing is required to address the three aforementioned differences. Underwater images require image enhancement, color balance, and noise removal [27]. There have been several breakthroughs in the field of underwater image processing [28,29,30,31]. Xin et al. [28] proposed a method based on the affine shadow transform and adaptive histogram equalization to achieve uniform illumination in a processed image. Ma et al. [29] proposed a method based on an affine shadow transform that reduces the color deviation of the processed image and improves its clarity. Zhang et al. [30] proposed a method based on biological principles that adaptively uniformizes the global brightness of an image according to its overall brightness distribution without manual intervention. Wang et al. [31] proposed a dual information modulation network that effectively supplemented the network’s structural and global perception capabilities by utilizing the differences and complementarities between features of different scales, thereby further improving the clarity and contrast of the image.

Previous research has played a significant role in the processing and optimization of underwater images. However, the optimization effect on the crack feature areas of underwater images is insignificant. Perhaps because of the different fields used in the research, most studies have only focused on optimizing the overall quality of underwater images and have neglected the optimization and denoising of specific areas (crack regions). Therefore, a more suitable underwater image-processing method must be developed to optimize underwater images with crack features. This study proposes a new underwater image-processing technology combining a novel white balance algorithm and bilateral filtering method to improve the image quality of processed underwater crack images, compared with previous image-processing methods.

After obtaining water crack images with underwater crack features, a crack detection network is used to achieve crack region detection. With the popularity of target detection networks, such as the YOLO (You Only Look Once) and U-net networks, digital image detection technology has become a detection method with a lower cost and higher efficiency. Moreover, it is widely used in underwater crack detection in concrete dams. Currently, there are many studies on the improvement and innovation of crack detection networks. Li et al. [32] proposed a new IDP-YOLOv8 network based on the newly launched YOLOv9 network, which utilizes a parallel architecture, including an image denoising and enhancement module as well as an improved YOLOv9 object detection module that can significantly improve image quality. Wu et al. [33] proposed an integrated approach based on the convolutional neural network DeepLabv3+ for crack detection as well as a pixel-level crack quantification algorithm that achieved high accuracy in crack detection. Abbasi et al. [34] proposed a fog perception adaptive YOLO algorithm for target detection in foggy weather environments, and this algorithm achieved a reasonable improvement in terms of average accuracy. Luo et al. [35] used a deep convolutional neural network for drone detection. In the process of feature extraction in convolutional layers of deep learning neural networks, there are often issues with insufficient and imprecise feature extraction, which can lead to decreased model accuracy. The OREPA convolutional structure addresses this problem. By integrating residual connections, edge information, and pyramid attention mechanisms, the OREPA convolutional layer enhances the model’s ability to extract complex features, thereby resolving the shortcomings of traditional convolutional layers in feature extraction and significantly improving the model’s recognition accuracy. A detailed introduction to the OREPA convolutional layer will be provided in Section 3.

Previous studies based on deep-learning networks have made significant progress in object detection [36]. A new deep-learning network, YOLOv9 [37], was proposed and used for object detection and segmentation. There has been relatively little research on the improvement of the YOLOv9 network; therefore, further research is required on the YOLOv9 network.

Building on previous research, this study proposes a novel image-processing technology to convert underwater crack images to water surface crack images and then replace the original convolutional blocks in YOLOv9 with OREPA convolutional blocks to improve the ability of the YOLOv9 network to accurately detect underwater cracks. The contributions of this study are as follows:(1)This study proposes a novel white balance algorithm that improves the image quality of processed underwater crack images, compared with previous white balance algorithms.(2)This study used a bilateral filtering denoising method to further denoise underwater crack images, reduce the impact of noise while preserving underwater crack features, and convert underwater crack images to water surface crack images. The denoising effect of bilateral filtering is superior to that of other denoising methods and can better restore the crack features of underwater images.(3)The original convolutional blocks of YOLOv9 were replaced with OREPA convolutional blocks. Compared with all the original models of YOLOv9, the proposed model shows better crack detection results in terms of the loss rate and other indicators, which applies to the method proposed in this study.

The remainder of this paper is organized as follows. Section 2 introduces the proposed underwater image-processing technology and various evaluation indicators for underwater images. Section 3 introduces the combination of the improved YOLOv9 model and the OREPA convolutional blocks. Section 4 describes the relevant experimental plans and results. Section 5 provides the conclusions. Section 5 provides the conclusions. Section 6 introduces the future work to be done and some limitations of this method.

## 2. A Novel Underwater Image-Processing Technology

This section introduces a novel white balance algorithm and the principle of bilateral filtering and then introduces the meaning and expressions of various indicators used to evaluate image quality.

### 2.1. A Novel White Balance Algorithm

White balance, also known as color constancy, is the ability to maintain color consistency in a scene under different lighting conditions. In an underwater environment, owing to different light source conditions, the light reflected from the same object may differ, leading to the final camera assuming that the same object has different colors under different lighting conditions. To address this issue, a white balance adjustment function is introduced in the image to increase or decrease the color values of the red, green, and blue channels at each pixel point, according to the perceived light, color, and temperature information. In this manner, the white objects in the image can be presented as truly white under different lighting conditions, and the other colors can maintain relative accuracy.

Four types of underwater image white balance algorithms are commonly used [38]: the average white balance method, perfect reflection method, image analysis-based chromatic aberration detection and correction method, and gray world white balance method. We introduce a new white balance method based on a dynamic threshold algorithm [39] to achieve image color uniformity.

This method introduces a new color space, namely, the YUV color space, where Y represents luminance and U and V represent chrominance (hue and saturation). The Y-component is constructed from the RGB input signals by combining specific portions of RGB signals. Chrominance is defined by two aspects of color: hue and saturation, which are represented by C_r_ and C_b_, respectively. Here, C_r_ reflects the difference between the red component of the RGB input signal and the luminance of the RGB signal, whereas C_b_ reflects the difference between the blue component of the RGB input signal and the luminance of the RGB signal.

This method determines a reference point by converting the RGB color space into the YC_r_C_b_ color space for analysis. By analyzing the YC_r_C_b_ coordinate space of the image, a region that is close to white and contains the reference point can be identified. A threshold is then set to designate certain points as reference points. Next, the input image is divided into 16 regions, and each region is transformed from the RGB color space to the YC_r_C_b_ space. A threshold is then set to determine the reference point position in each region. In Figure 2, the thirteenth region is used as an example.

The specific algorithm process is as follows.


(1)Transform the image from the RGB space into the YC_r_C_b_ space.
(1)$$Y=0.257R+0.054G+0.098B+\frac{16}{255}$$
(2)$$C_{b}=-0.148R-0.291G+0.439B+\frac{128}{255}$$
(3)$$C_{r}=-0.439R-0.368G-0.071B+\frac{128}{255}$$(2)Calculate the mean values of *C_r_* and *C_b_* for each region, where *M_r_*, *M_b_*, and *N* are the numbers of pixels in each region, and (*i*, *j*) denotes the reference point coordinates for each region.
(4)$$M_{r}=\frac{\sum C_{r}\left(i,j\right)\;}{N}\;,\;M_{b}=\frac{\sum C_{b}\left(i,J\right)}{N}$$
(5)$$D_{r}=\frac{\sum \left|C_{r}\left(i,\;j\right)-\left.M_{r}\right|\;\right.}{N}\;,\;D_{b}=\frac{\sum \left|C_{b}\left(i,\;j\right)-\left.M_{b}\right|\;\right.}{N}\;$$(3)Find a reference point that satisfies the conditions in each region. The judgment criteria are shown in Equation (6).
(6)$$D_{r\;}=\geq 0.01;\;D_{b\;}\geq 0.01$$(4)After determining the reference point position, calculate the maximum brightness *Y_max_* of the reference point.(5)Calculate the gain coefficients (*R_gain_*, *G_gain_*, and *B_gain_*) of each RGB channel. In Equation (7), *R_avgw_*, *G_avgw_*, and *B_avgw_* are the mean values of the reference points for the three channels in each region.
(7)$$R_{gain}=\frac{Y_{max}}{R_{avgw}}\;,\;G_{gain\;}=\frac{Y_{max}}{G_{avgw}}\;,\;B_{gain}=\frac{Y_{max}}{B_{avgw}}$$(6)Calculate the gain coefficients (*R_gain_*, *G_gain_*, and *B_gain_*) for each RGB channel. In Equation (7), *R_avgw_*, *G_avgw_*, and *B_avgw_* are the mean values of the reference points for the three channels in each region.(7)The RGB channel value of the image is calculated using the gain coefficient. In Equation (8), *R*′, *G*, and *B*′ are the RGB channel values of the new image calculated by the gain coefficient in each region.
(8)$$R^{\prime \;}=R\times R_{gain\;},\;G^{\prime }=G\times G_{gain}\;,\;B^{\prime }=B\times B_{gain}$$(8)Combine the newly obtained RGB channel (*R*′, *G*′, and *B*′) values to generate new underwater crack images. Figure 3 shows different underwater crack images generated by the four classic white balance algorithms and the white balance algorithm proposed in this study.


### 2.2. Bilateral Filtering Denoising

Image denoising has many common methods, such as median filtering, Gaussian filtering, and mean filtering. This paper uses the bilateral filtering denoising method to process crack feature images [40,41]. Bilateral filtering is a non-linear filtering method that combines the spatial proximity and pixel similarity of an image, taking into account both spatial information and grayscale similarity to achieve edge preservation and denoising. The advantage of the bilateral filter is that it can preserve edges. Usually, Gaussian filtering is used for denoising, which blurs the edges more and does not have a significant protection effect for high-frequency details. The reason why the bilateral filter can achieve smooth denoising while preserving the edges well is that the filter kernel is generated by two functions: the spatial domain kernel and the value domain kernel. In the spatial domain kernel, the bilateral filter follows Equation (9). The ω_d_ term is a template weight determined by the Euclidean distance between pixel positions, *q*(*i*, *j*) is the coordinates of other coefficients in the template window, *p*(*k*, *l*) is the center coordinate point of the template window, and *σ_d_* is the standard deviation of the Gaussian kernel function in the spatial domain and is used to control the weights of pixel positions.
(9)$$\omega_{d}\left(i,\;j,\;k,\;l\right)=\exp(-\frac{{\left(i-k\right)}^{2}-{\left(j-l\right)}^{2}}{{2\sigma }_{d}^{2}})\;$$

Within the range kernel, bilateral filtering is as follows in Equation (10). Here, *ω_r_* is the template weight determined by the difference in pixel values, *f*(*i*, *j*) represents the pixel value of the image at point *q*(*i*, *j*), *f*(*k*, *l*) denotes the pixel value of the image at point *p*(*k*, *l*), and *σ_r_* is the standard deviation of the Gaussian kernel function in the pixel value domain that is used to control the weights of pixel values.
(10)$$\omega_{r}\left(i,\;j,\;k,\;l\right)=\exp(-\frac{\left\|f\left(i,j\right)-\left.f\left(k,l\right)\right\|\right.}{{2\sigma }_{r}^{2}})\;$$

Here, *ω_d_* measures the distance between two points, with lower weights as the distance increases, and *ω_r_* measures the degree of similarity in pixel values between two points, with higher weights assigned to points that are more similar. Finally, multiplying the above two templates yields the template weights *ω* for the bilateral filter.
(11)$$\omega \left(i,\;j,\;k,\;l\right)=e\mathrm{xp}(-\frac{{\left(i-k\right)}^{2}-{\left(j-l\right)}^{2}}{{2\sigma }_{d}^{2}})-(-\frac{\left\|f\left(i,j\right)-\left.f\left(k,l\right)\right\|\right.}{{2\sigma }_{r}^{2}})\;$$

The data equation for the bilateral filter is given in Equation (12):(12)$$g\left(i,\;j\right)=\frac{\sum_{kl}f\left(k,\;l\right)\omega \left(i,\;j,k,l\right)}{\sum_{kl}\omega \left(i,j,k,l\right)}$$

This study uses median filtering, mean filtering, Gaussian filtering, and bilateral filtering to denoise the input image and then compares the values of various evaluation indicators of the processed image to find the most suitable denoising method. Figure 4 illustrates the specific process.

### 2.3. Image Evaluation Indicators

The main image metrics used in this study were the UIQM [23], SSIM, and PSNR [42]. UIQM includes three underwater image attribute metrics: the underwater image color metric (UICM), the underwater image clarity metric (UISM), and the underwater image contrast metric (UIConM). Below are the methods used to calculate various evaluation indicators. The solution equations for the UICM, UISM, and UIConM terms in Equation (13) can be found in the literature [23].
(13)UIQM=0.0282×UICM+0.2953×UISM+3.5753×UIConM

Given a reference image *f* and a test image *g*, both of size *M* × *N*, the PSNR between *f* and *g* is defined as follows:(14)$$\mathrm{PSNR}\left(f,g\right)=10\log_{10}\left(\frac{255^{2}}{\mathrm{MSE}\left(f,g\right)}\right)$$
(15)$$\mathrm{MSE}\left(f,g\right)=\frac{1}{MN}\;\sum_{i=1}^{M}\sum_{j=1}^{N}(f_{ij}-g_{ij}{)}^{2}$$

The PSNR value approaches infinity as the MSE [42] approaches zero, indicating that a higher PSNR value provides a higher image quality. On the other end of the scale, a small PSNR value implies large numerical differences between the images.

The SSIM is a quality metric used to measure the similarity between two images. This is considered to correlate with the quality perception of the human visual system. The SSIM is defined as follows:(16)$$\mathrm{SSIM}\left(f,g\right)=l\left(f,g\right)c\left(f,g\right)s(f,g)$$
(17)$$\left\{\begin{array}{c} l\left(f,g\right)=\frac{{2\mu }_{f}\mu_{g}+C_{1}}{\mu_{f}^{2}+\mu_{g}^{2}+C_{1}}\\ c\left(f,g\right)=\frac{{2\sigma }_{f}\sigma_{g}+C_{2}}{\sigma_{f}^{2}+\sigma_{g}^{2}+C_{2}}\\ S\left(f,g\right)=\frac{\sigma_{fg\;}+C_{3}}{\sigma_{f}\sigma_{g}+C_{3}} \end{array}\right.$$

Here, *l*(*f*, *g*) is used to measure the closeness of the average brightness (*μ_f_* and *μ_g_*) between two images, *c*(*f*, *g*) is used to measure the closeness of contrast (*σ_f_* and *σ_g_*) between two images, and *s*(*f*, *g*) is used to measure the correlation coefficient between the images *f* and *g*. After describing the UIQM, PSNR, and SSIM, this study compares various underwater image-processing methods using these three indicators and thus determines the most suitable underwater image-processing method.

### 2.4. A New Underwater Image-Processing System

This study proposes a novel underwater image-processing technology combined with bilateral filtering and a novel white balance algorithm for underwater image denoising to achieve image enhancement and compensation. This method is used to convert underwater crack images into above-water crack images. Figure 5 shows a detailed structural diagram. Box 1 in Figure 5 represents the five white balance algorithms from top to bottom: the average white balance method, perfect reflection method, proposed white balance method, chromatic aberration detection and correction method based on image analysis, and gray world method. Box 2 in Figure 5 shows the four filtering and denoising methods from top to bottom: mean, median, bilateral, and Gaussian filtering. The results of the experiment described in Section 4.1 indicate that the proposed white balance method and bilateral filtering denoising method have significant advantages over other methods in processing underwater images, and can convert underwater crack images into above-water crack images without damaging the crack characteristics.

## 3. Improved Crack Detection Network Based on YOLOv9

### 3.1. Infrastructure of YOLOv9

YOLOv9 is a recently launched object detection network [37]. Figure 6 shows a structural diagram of the YOLOv9 network. The basic architecture of YOLOv9 is as follows. Many existing deep learning networks overlook the fact that a large amount of information is lost when the input data undergo feature extraction and spatial transformation, layer by layer. YOLOv9 delves into important data loss issues during data transmission in deep networks, namely, information bottlenecks and reversible functions. Moreover, it proposes the concept of programmable gradient information (PGI) [43] to address the various changes required for deep networks to achieve multiple objectives. PGI [44] can provide complete input information for the target task to calculate the objective function and thereby obtain reliable gradient information to update the network weights. In addition, a new lightweight network structure based on gradient path planning, namely, the generalized efficient layer aggregation network (GELAN) [45], was designed. The architecture of GELAN confirmed that the PGI achieved outstanding results with lightweight models.

The concept of PGI in YOLOv9 and the introduction of GELAN can help solve the feedforward problem of input data caused by information bottlenecks and reversible functions as well as reduce training losses.

The PGI mainly consists of three parts: the main branch, the auxiliary reversible branch, and multilevel auxiliary information. During the inference process, only the main branch was used without additional inference costs, whereas the other two branches were mainly used to solve information bottleneck problems. Figure 7 shows the overall architecture of the PGI.

The GELAN network combines the design of CSPNeT [46] and ELAN [45] by adopting the concepts of segmentation and reassembly from CSPNeT and introducing the hierarchical convolution processing from ELAN in each section. The design of GELAN makes the overall training model lightweight with a higher inference efficiency and accuracy. Figure 8 shows the structural diagrams of the three network architectures.

### 3.2. Improved YOLOv9 Network Combined with an OREPA Module

OREPA (Online Convolutional Re-parameterization) reduces the cost and complexity of training deep learning models via online convolution reparameterization. This method includes two stages. First, a special linear scaling layer is used to optimize the performance of the online block. Second, the training overhead can be reduced by compressing complex training modules into a single convolution. The OREPA process includes three main steps: removal, scaling, and normalization. After a series of steps, the OREPA method effectively simplifies the complexity of the model and optimizes the training efficiency during the training phase. Figure 9 illustrates the specific steps of the OREPA process.

A key component of OREPA is the linear scaling layer, which aims to maintain complexity management during training while improving the flexibility and adaptability of the model in learning features. The appropriate scaling of the weights helps stabilize the entire training process and enables the model to converge as quickly as possible. The linear scaling layer consists of the following four parts: a frequency prior filter, linear depth-separable convolution, 1 × 1 convolution with heavy parameters, and linear deep stem cells [47]. Figure 10 shows the four components of the OREPA.

During the model training process, the OREPA framework uses module compression to reduce memory usage and computational costs. The design of OREPA allows multiple convolutional layers and batch normalization layers to be merged into one convolutional layer via reparameterization, thereby reducing the training complexity and memory requirements. Figure 11 shows a comparison of the different convolutional layers during training.

The improved YOLOv9-OREPA model proposed in this study uses OREPA convolution layers instead of the original YOLOv9 network’s convolutional layers to reduce memory usage, but computational costs are incurred during model training to improve training accuracy. Figure 12 presents a flowchart of the YOLOv9-OREPA model. Figure 6 and Figure 12 together show that the difference from the original YOLOv9 structure lies in the replacement of the convolutional structure.

### 3.3. Loss Function and Evaluation Metrics of YOLOv9-OREPA

The loss function of YOLOv9-OREPA consists of two parts: classification loss (*Loss_cla_*) and regression loss (*Loss_box_*), as follows.
(18)$$Loss={Loss}_{cla}+{Loss}_{box}$$

The classification loss adopts binary cross-entropy (*BCE*), which is used to determine whether the detection target belongs to a particular category, and accordingly outputs the following network confidence result:(19)$${Loss}_{cla-BCE}=\frac{1}{N}\sum_{i}-\left[y_{i}.\log\left(p_{i}\right)+\left(1-y_{i}\right).\log\left(1-p_{i}\right)\right]$$
where *y_i_* represents the target category and *p_i_* represents the probability that the predicted result is *y_i_*. In this article, there is only one classification category (cracks), so the classification loss can be ignored. Regression loss is caused by the difference in coordinates and dimensions between the predicted and target boxes. In the regression tasks, the degree of regression can be measured using the ratio of the predicted box to the target box. The IoU ratio (*IoU*) is a descriptive method expressed via the following formula, where *A* is the prediction and *B* is the target, as follows:(20)$$IoU=\frac{\left|A+B-A\cup B\right|}{\left|A\cup B\right|}$$

This study uses *CIoU* as the regression loss function, adds the aspect ratio to the basic *IoU*, and considers three factors, namely, the overlap area, center point distance, and aspect ratio of the box, to improve the regression accuracy [48]. Figure 13 illustrates the calculation principle of the loss function.
(21)$$L_{CIoU}=1-IoU+\frac{\rho^{2}\left(b,b^{gt}\right)}{c^{2}}+\alpha v$$
(22)$$c^{2}=(w^{c}{)}^{2}+(h^{2}{)}^{2}$$
(23)$$\alpha =\frac{v}{\left(1-IoU\right)+v^{2}}$$
(24)$$v=\frac{4}{\pi^{2}}(arctan\frac{w^{gt}}{h^{gt}}-arctan\frac{w}{h}{)}^{2}$$

The final loss function is as follows:(25)$$Loss={Loss}_{cla-\mathrm{BCE}}+{Loss}_{box-CIoU}$$

The performance of the improved YOLOV9-OREPA model was evaluated using object detection evaluation metrics, including the accuracy *P*, recall *R*, F_1_-value *F*_1_, and average category accuracy *mAP*. The specific calculation method is as follows:(26)$$P=\frac{TP}{TP+FP}$$
(27)$$R=\frac{TP}{TP+FN}$$
(28)$$F_{1}=\frac{2\times P\times R}{P+R}$$
(29)$$AP=\int_{0}^{1}P\left(R\right)dR$$
(30)$$mAP=\frac{\sum_{q=1}^{Q}AP(q)}{Q}$$
where *TP* is the number of correctly detected samples, *FP* is the number of incorrectly detected samples, *FN* is the number of undetected defect samples, *F*_1_ is the harmonic average of the accuracy *P* and recall rate *R*, the single-category *AP* is the area of the closed graph formed by the curve of the accuracy *P* and recall rate *R*, the coordinate axis *Q* represents the category, and the average category average precision *mAP* is the average single-category *AP* of each of the defect categories.

## 4. Case Study

### 4.1. Underwater Image Optimization Experiment

To obtain underwater images with the crack features required for the experiment, an underwater robot was used to capture images in a laboratory swimming pool. Underwater images were captured under low-light conditions to verify the superiority and reliability of the proposed method. Figure 14 shows the experimental setup used. Figure 15 shows the experimental process.

After capturing multiple images, five images in Figure 16 with obvious crack features were selected for use in the experiment. Subsequently, a white balance operation was performed on the five images. The initial image in Figure 16a was considered as an example. Five methods were used: the average white balance method (first method), perfect reflection method (second method), our method (third method), chromatic aberration detection and color correction method based on image analysis (fourth method), and gray world method (fifth method). In Figure 17, five processed images were obtained, and the UIQM, PSNR, and SSIM of each image were compared to determine the best white balance method.

The image metrics obtained from the first image using the different white-balance methods were different, as shown in Table 1.

Table 1 shows that the proposed method exhibits good processing performance in terms of the UIQM, PSNR, and SSIM evaluation indicators. Our proposed method increases the UICM index of the initial underwater image from 14.897 to 18.395, increases the UISM index from 0.136 to 0.140, decreases the UIConM index from −0.119 to −0.118, and increases the UIQM index from 0.032 to 0.136. The PSNR index is the highest for the proposed method, reaching 39.783, and the SSIM index is also the highest for the proposed method, reaching 0.988. Hence, this method not only retains the crack features of the original image but also enhances and compensates the original image. Figure 18 shows the processing and data tables for the remaining images.

Table 2 shows that in the processing of the five underwater images, our method performs better than the other four methods in terms of three image evaluation indicators (i.e., UIQM, SSIM, and PSNR). Especially in terms of the UISM evaluation index, our proposed white balance method has the highest gain in image quality, improving Figure 16a–e from the original (0.032, −0.063, 0.082, 0.094, 0.069) to (0.136, 0.051, 0.084, 0.112, 0.112). Therefore, it can be concluded that among the many white-balance processing methods, our method has the best processing effect on underwater images. The five images processed using our white balance method were the output results of the first step of the proposed method. Next, the five processed images were denoised, and these images were used as the input images for the second step.

The underwater images processed by our white balance method are the output images of the first step and the input images of the second-step denoising operation. Considering Figure 19a as an example, Figure 20 presents a comparison of the denoising processes. First, Figure 19a was used as the input image, and four different denoising methods (i.e., bilateral, median, mean, and Gaussian filtering) were used to denoise the image. The UIQM, PSNR, and SSIM of the denoised images were compared to determine the most suitable denoising method.

The other input images in Figure 19 also underwent various denoising operations. In the denoising experiments of mean filtering and Gaussian filtering, different convolution kernel sizes were added to denoise the image, and the various evaluation indicators of the processed images were compared to determine the denoising method that best suited the underwater image. Table 3 shows the results.

Table 3 shows that the UIQM and PSNR image metrics of the five images used in the experiment were the best among all denoising methods after bilateral filtering denoising. According to the definition of the SSIM, image denoising affects image similarity. The bilateral filtering method impacts the fourth decimal place, satisfying the accuracy requirements. In addition, the results of the Gaussian and mean filtering experiments indicate that the larger the convolution kernel used, the larger the UIQM of the image, whereas the PSNR and SSIM metrics are smaller. This is because the larger the convolution kernel used for denoising, the less noise in the image, the smoother the image itself, and the larger the UIQM metric. However, the denoising operation may cause a loss of crack features. Therefore, the PSNR and SSIM metrics are slightly smaller than those of the input image.

The new image-processing method proposed in this study combines a novel white balance method and a bilateral filtering denoising method and optimizes three of the image metrics used in the experimentation. Hence, it is suitable for underwater image processing and makes it possible to convert underwater crack images into above-water crack images that preserve crack features. Table 4 summarizes the overall improvements in the image metrics of the five experimental images processed using the proposed method. The UIQM index of the image processed by this method is significantly improved, whereas the PSNR and SSIM indexes also reach an acceptable level, indicating that this method not only retains the characteristics of the original image but also confers a significant improvement in image quality, thereby exceeding the performance of than other methods. In summary, the method proposed in this study is a new image-processing method suitable for underwater images.

### 4.2. Improved YOLOv9-OREPA Network Performance Evaluation

This study used 938 images with crack information as the dataset and divided the image dataset into training, validation, and test sets during the training process. The test set was not used for training. The training set consisted of 750 images, whereas the validation and test sets consisted of 96 images each, at a ratio of 8:1:1. Table 5 lists the basic operating parameters of this model.

To verify that the proposed YOLOv9-OREPA has improved performance compared with the original YOLOv9, this study compares all the original YOLOv9 models (i.e., YOlOv9, YOlOv9-e, YOlOv9-c, YOlOv9-m, YOlOv9-s, and YOlOv9-t) with the YOLOv9-OREPA model. Table 6 shows the results.

Table 6 shows that the performance of YOLOv9-OREPA proposed in this study exceeds those of various existing YOLOv9 models. Compared with the initial YOLOv9, *R* increased by 5.5%, *F*_1_ increased by 2%, *mAP*_0.5_ increased by 0.9%, and *mAP*_0.5:0.95_ increased by 2.1%. This proves that OREPA, a new convolution mode, is highly suitable for improving YOLOv9.

Figure 21a,b show that after 200 rounds of training, YOLOv9-OREPA had the smallest training and validation process losses, with a training process loss of 1.39% and a validation process loss of 1.10%. In addition, Figure 21c,d show that YOLOv9-OREPA has a better convergence trend than do the other YOLOv9 models, with relatively stable fluctuations in the *P* and *R* curves and a continuous upward trend. Furthermore, Figure 21e,f show that YOLOv9-OREPA is also superior to other ordinary models in terms of the average category accuracy *mAP*. YOLOv9-OREPA had the highest *mAP*_0.5_ (74.8%) and *mAP*_0.5:0.95_ (44.1%).

The precision–recall curves for the accuracy and summoning rate visually demonstrate the performance of the model. Plotting the precision–recall curves of the different models in Figure 22a reveals that the closed graph area enclosed by the YOLOv9-OREPA curve and coordinate axis is the largest, which can intuitively demonstrate the higher performance of the improved YOLOv9-OREPA. Moreover, Figure 22b shows that the *F*_1_ curve of YOLOv9-OREPA has the largest area with the coordinate axis; when the confidence interval is 0.357, the *F*_1_ value is the highest, reaching 0.73.

Figure 23 shows the crack detection results of the different original YOLOv9 models and the YOLOv9-OREPA model proposed in this study after training. The results show that the base YOLOv9 model exhibits reinspection and missed detection in the crack detection process, while the improved YOLOv9-OREPA model does not have reinspection or missed detection issues indicating that the improved YOLOv9-OREPA model enhances the accuracy of crack detection.

The YOLOv9-OREPA model can achieve the goal of crack detection, and its performance is superior to those of other YOLOv9 models.

Figure 24 shows the crack detection results of the original underwater image after combining the new underwater image-processing method proposed in this study with the improved YOLOv9-OREPA algorithm. After image processing, the overall indicators of the image are improved while ensuring the crack features. The proposed YOLOv9-OREPA network could fully perform crack detection. The images processed using the underwater image-processing technology proposed in this study can be accurately identified by the improved YOLOv9 OREPA network trained on underwater crack images to thereby complete the conversion from underwater to above-water crack images.

Figure 25 shows that the proposed method is suitable for crack detection in not only underwater images under low-light conditions but also underwater and above-water images with large color differences, which proves the superiority and universality of this method in crack detection.

## 5. Conclusions

This study proposes a novel underwater image-processing method that combines a new white balance technique with bilateral filtering denoising. The metrics of the processed underwater crack images were evaluated and compared with those of other methods, demonstrating superior performance. These results further demonstrate that the proposed underwater image-processing method can convert underwater crack images into high-quality images that retain the original crack features to thereby transform underwater crack images into above-water crack images. Subsequently, crack detection was conducted based on the improved YOLOv9-OREPA model, and comparisons were made between the improved YOLOv9-OREPA model and other basic YOLOv9 models. The experimental results indicate that YOLOv9-OREPA significantly improves the detection metrics of YOLOv9. In summary, the following conclusions were drawn:(1)In a comparative experiment of white balance algorithms, the proposed novel method was superior to the other four tested white balance methods in terms of the image UIQM, PSNR, and SSIM indices, which proves the superiority of our proposed method.(2)In the comparative experiment of the denoising effect, the bilateral filtering method was superior to the other denoising methods in the evaluation of various image indicators, thus proving the superiority of bilateral filtering in denoising underwater crack images. The innovative image-processing technique proposed in this study has significant advantages and can greatly improve various underwater image metrics to transform underwater images into above-water images.(3)The results of a comparative experiment on crack detection between YOLOv9-OREPA and the YOLOv9 series show that YOLOv9-OREPA improves various training indicators of crack detection and is a deep-learning network that is more suitable for crack detection.

In summary, this study innovates from the two aspects of underwater image processing and crack detection network optimization. The system proposed in this study is better aligned with the requirements of underwater crack detection, improves the image quality of underwater crack images and the accuracy of crack detection, and converts underwater crack images into above-water crack images. This study offers a new perspective on underwater crack detection by transforming underwater images into above-water images, thereby addressing the issue of insufficient datasets that can lead to imprecise underwater crack detection.

## 6. Future Works

The input images used in this study were all collected from a laboratory water tank, which differs from the crack areas in real reservoir dam environments. The water in the laboratory tank is relatively clear, which also differs significantly from the water quality conditions in actual environments. In the future, we plan to collect crack feature images in real environments using underwater robots at reservoirs and apply the image enhancement and crack detection methods proposed in this paper to validate their accuracy and superiority. Additionally, we will create a murky water environment in the laboratory to extract crack images and verify the general applicability of the method proposed in this paper.

## Figures and Tables

**Figure 2 sensors-24-05981-f002:**
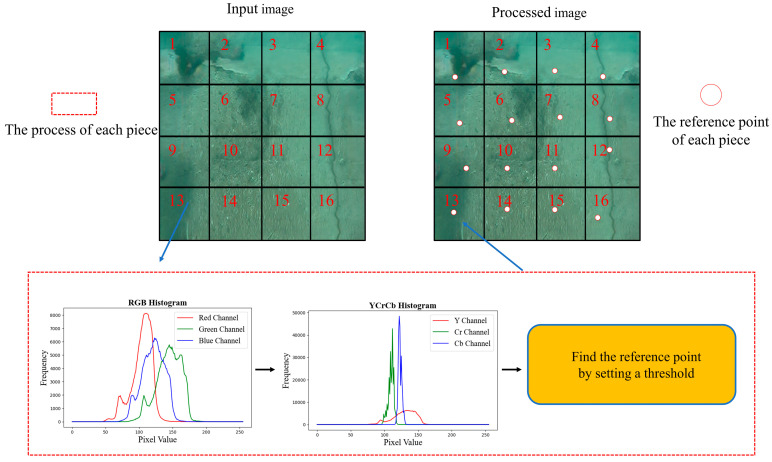
Principle of the new white balance method.

**Figure 3 sensors-24-05981-f003:**
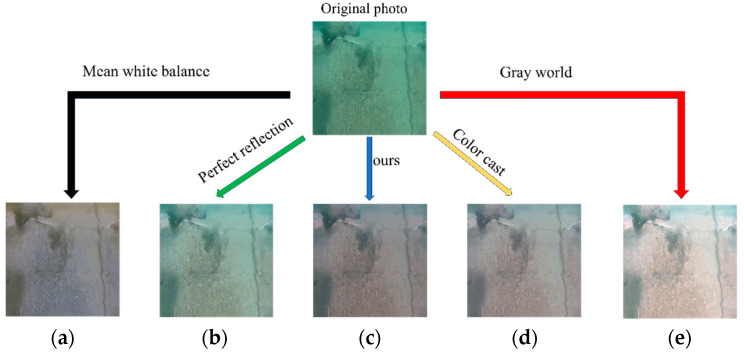
Five white balance algorithms: (**a**) mean white balance; (**b**) perfect reflection; (**c**) ours; (**d**) color cast; (**e**) gray world.

**Figure 4 sensors-24-05981-f004:**
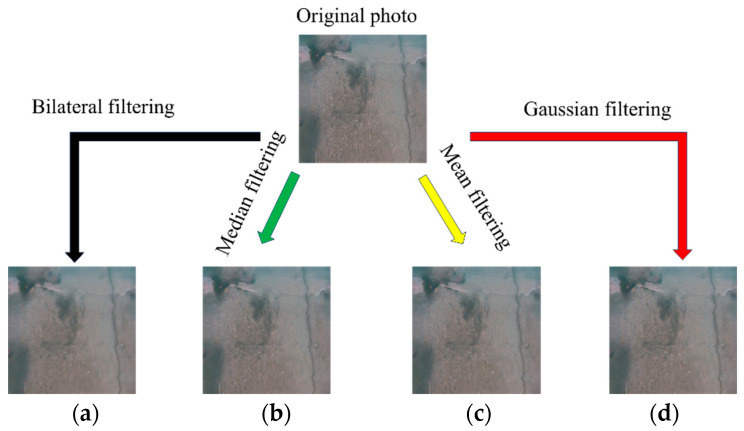
Four denoising methods: (**a**) bilateral filtering; (**b**) median filtering; (**c**) mean filtering; (**d**) Gaussian filtering.

**Figure 5 sensors-24-05981-f005:**
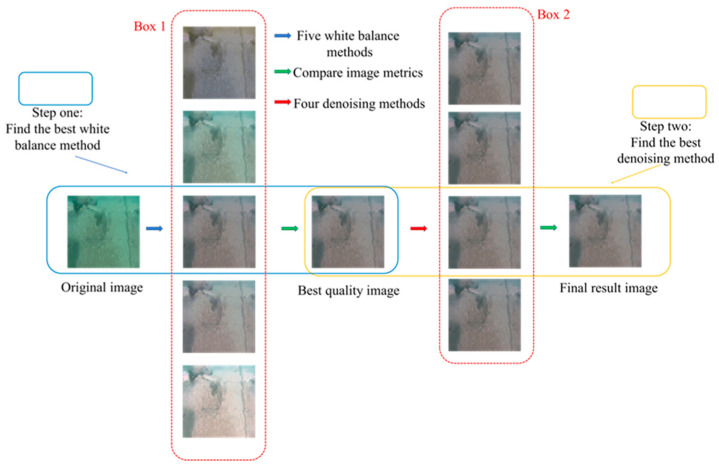
New underwater image-processing workflow.

**Figure 6 sensors-24-05981-f006:**
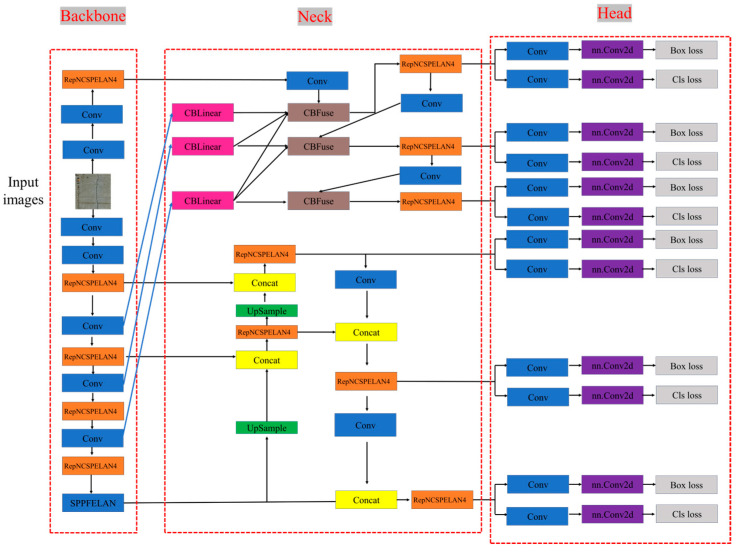
Infrastructure diagram of YOLOv9.

**Figure 7 sensors-24-05981-f007:**
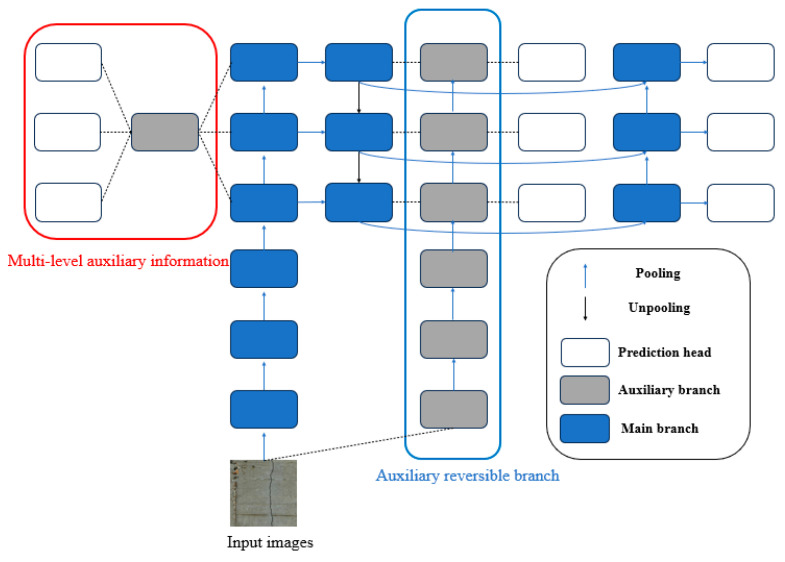
Overall architecture of PGI.

**Figure 8 sensors-24-05981-f008:**
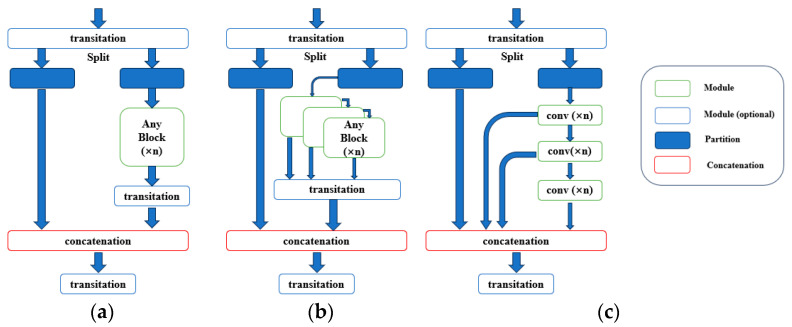
Structure diagrams of three network architectures: (**a**) CSPNeT; (**b**) GELAN; (**c**) ELAN.

**Figure 9 sensors-24-05981-f009:**
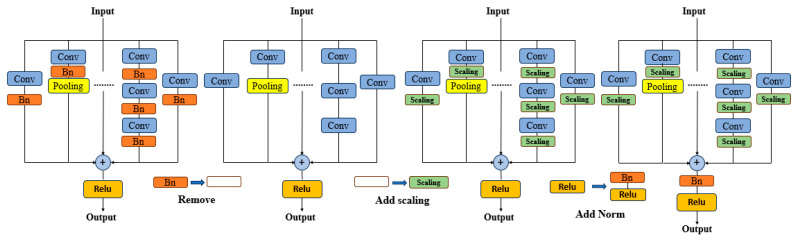
Specific steps of the OREPA process.

**Figure 10 sensors-24-05981-f010:**
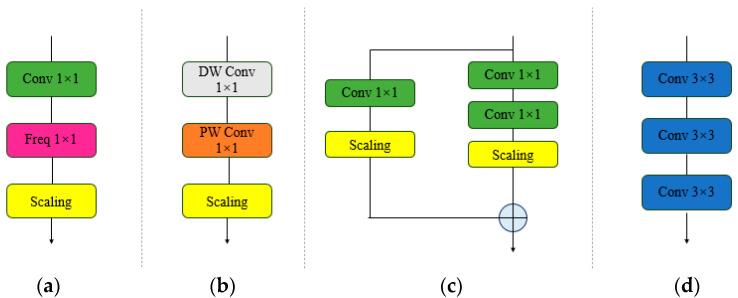
Four components proposed in OREPA: (**a**) frequency prior filter; (**b**) linear depth separable convolution; (**c**) 1 × 1 convolution with heavy parameters; (**d**) linear deep stem cells.

**Figure 11 sensors-24-05981-f011:**
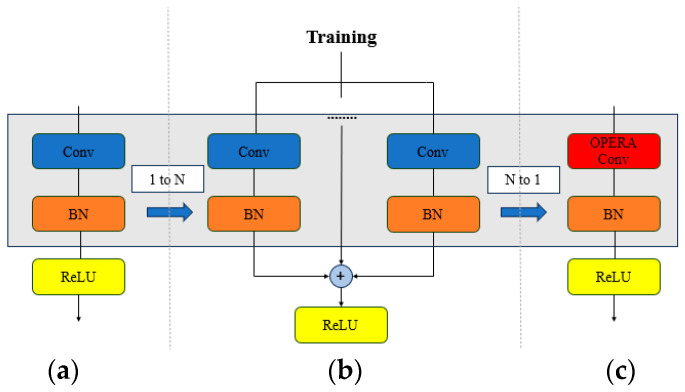
Comparison of different convolutional layers during training: (**a**) standard convolutional layer (without intermediate parameterization); (**b**) typical reparameterization; (**c**) OREPA.

**Figure 12 sensors-24-05981-f012:**
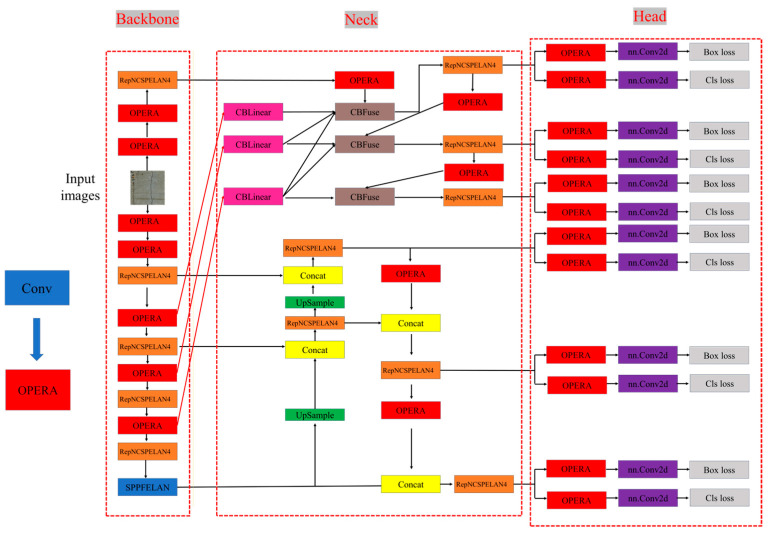
Infrastructure diagram of the YOLOv9-OREPA model.

**Figure 13 sensors-24-05981-f013:**
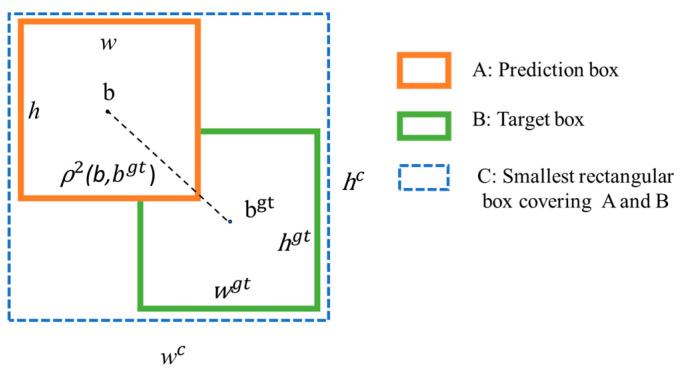
Calculation principle of the loss function.

**Figure 14 sensors-24-05981-f014:**
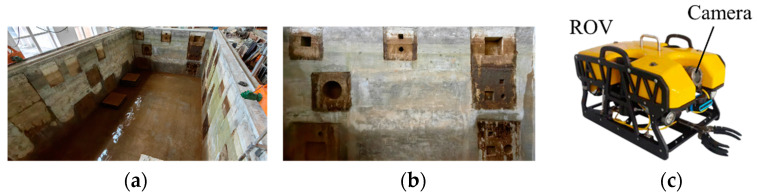
Experimental facilities: (**a**) laboratory pool; (**b**) crack defect wall; (**c**) underwater robots used.

**Figure 15 sensors-24-05981-f015:**
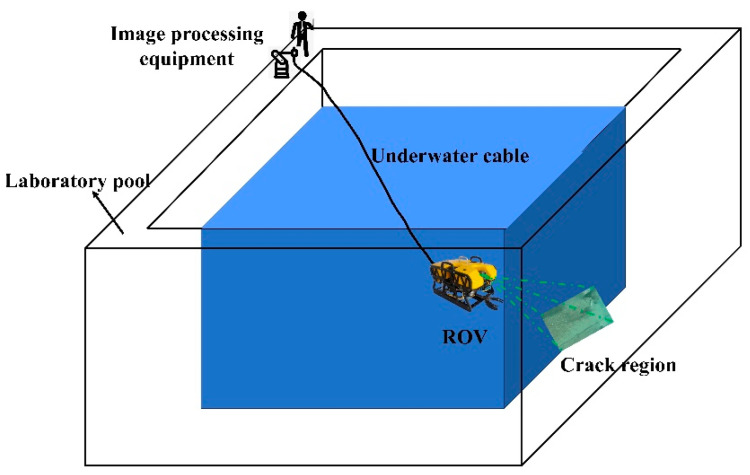
Experimental process image.

**Figure 16 sensors-24-05981-f016:**
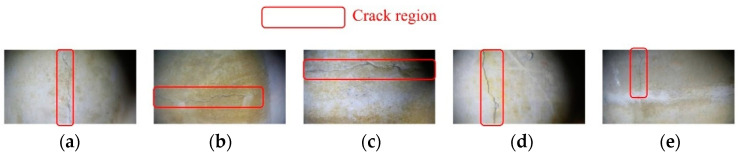
Five underwater images with crack features under low light conditions.

**Figure 17 sensors-24-05981-f017:**
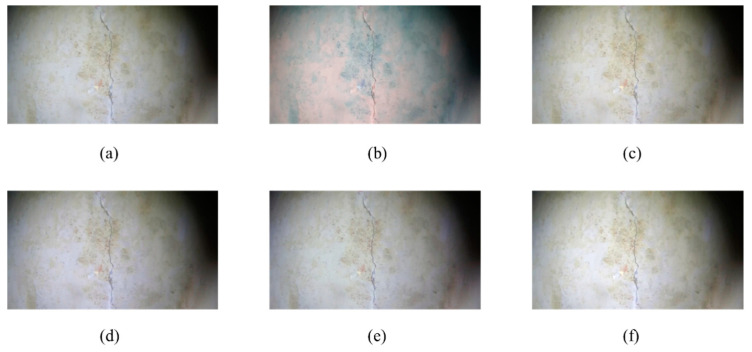
Image white balance operations: (**a**) original image; (**b**) mean white balance; (**c**) perfect reflection; (**d**) ours; (**e**) color cast; (**f**) gray world.

**Figure 18 sensors-24-05981-f018:**
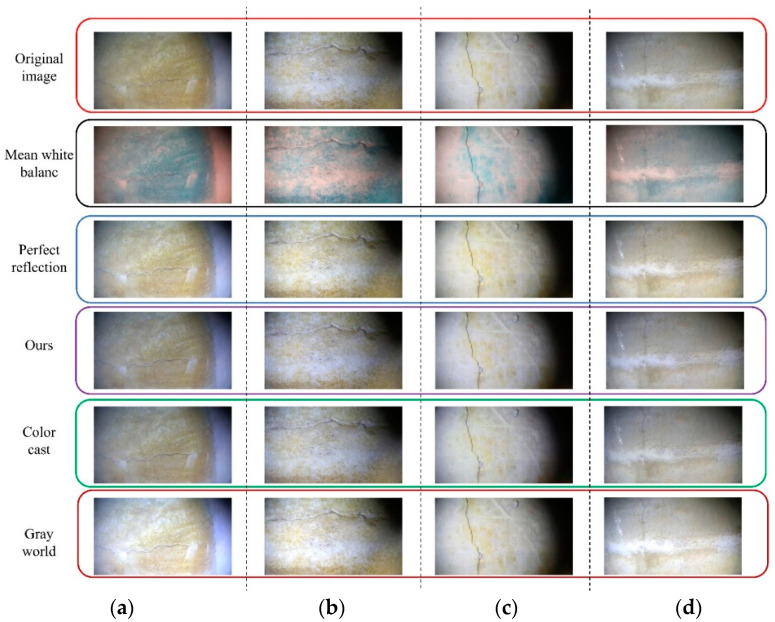
The white balance processing results of the remaining four original images: (**a**) processing results of Figure 16b; (**b**) processing results of Figure 16c; (**c**) processing results of Figure 16d; (**d**) processing results of Figure 16e.

**Figure 19 sensors-24-05981-f019:**
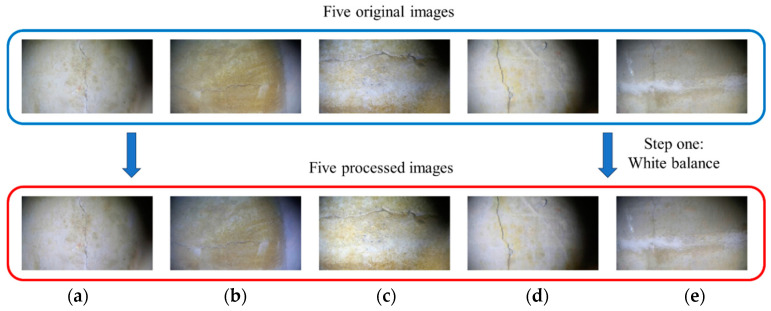
Images processed via our proposed method: (**a**) original picture from Figure 16a; (**b**) original picture from Figure 16b; (**c**) original picture from Figure 16c; (**d**) original picture from Figure 16d; (**e**) original picture from Figure 16e.

**Figure 20 sensors-24-05981-f020:**
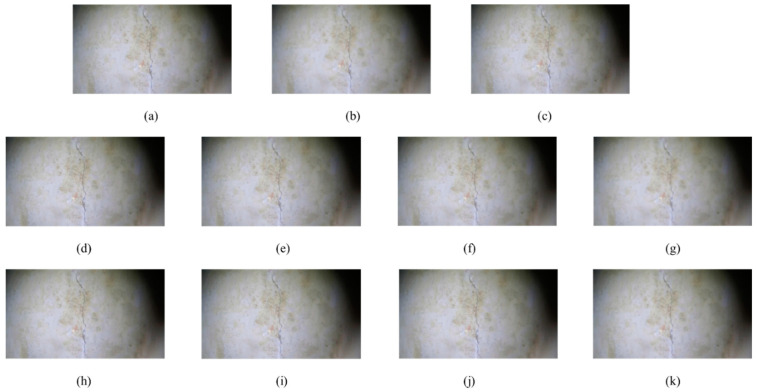
Image denoising process diagram: (**a**) input image; (**b**) bilateral filtering; (**c**) median filtering; (**d**) mean filtering (1 × 1 convolutional kernel); (**e**) mean filtering (3 × 3 convolutional kernel); (**f**) mean filtering (5 × 5 convolutional kernel); (**g**) mean filtering (7 × 7 convolutional kernel); (**h**) Gaussian filtering (1 × 1 convolutional kernel); (**i**) Gaussian filtering (3 × 3 convolutional kernel); (**j**) Gaussian filtering (5 × 5 convolutional kernel); (**k**) Gaussian filtering (7 × 7 convolutional kernel).

**Figure 21 sensors-24-05981-f021:**
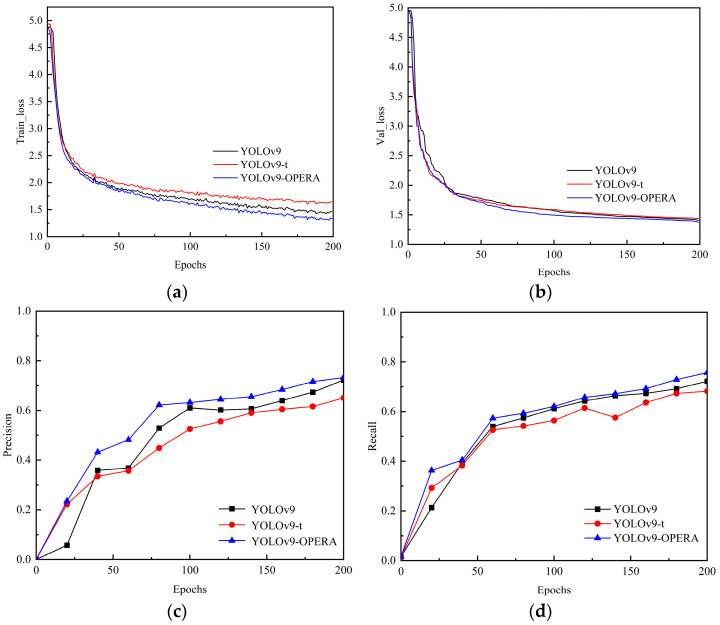
Various performance indicators of different models: (**a**) Train_loss; (**b**) Val_loss; (**c**) precision; (**d**) recall; (**e**) *mAP*_0.5_; (**f**) *mAP*_0.5:0.95_.

**Figure 22 sensors-24-05981-f022:**
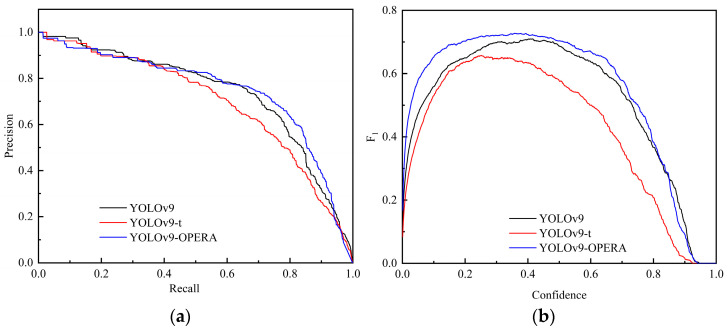
Other performance indicators of different models: (**a**) the precision–recall curve; (**b**) *F*_1_ curve.

**Figure 23 sensors-24-05981-f023:**
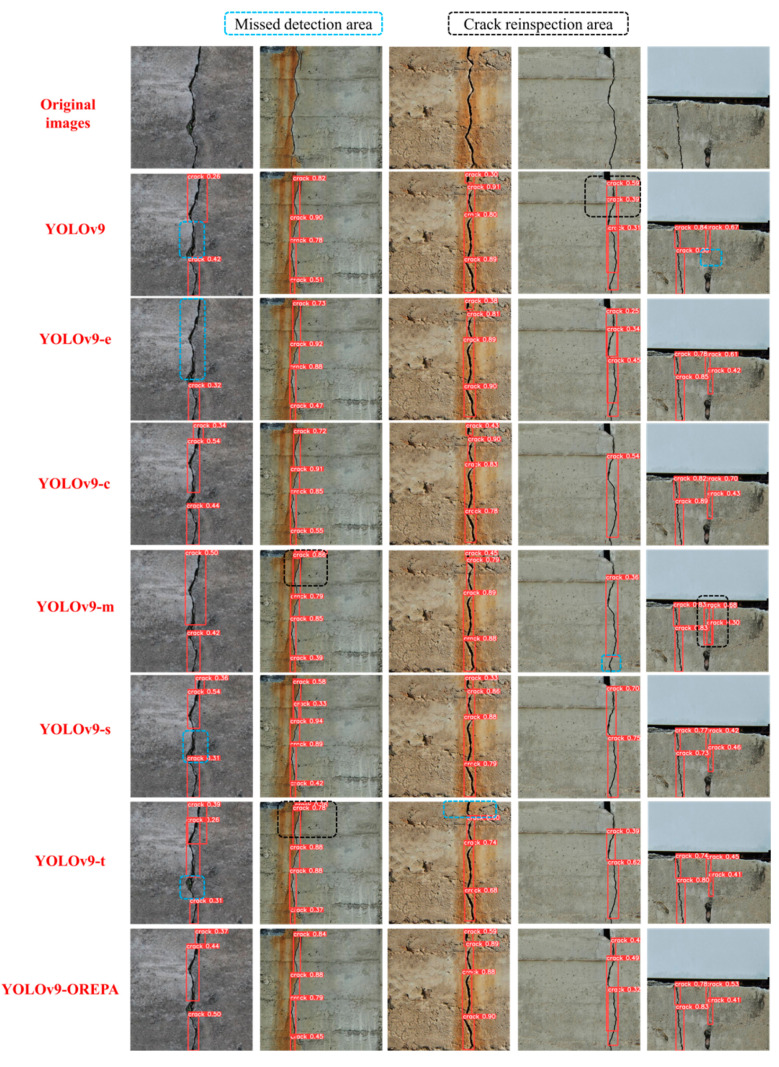
Crack detection results of different YOLOv9 models.

**Figure 24 sensors-24-05981-f024:**
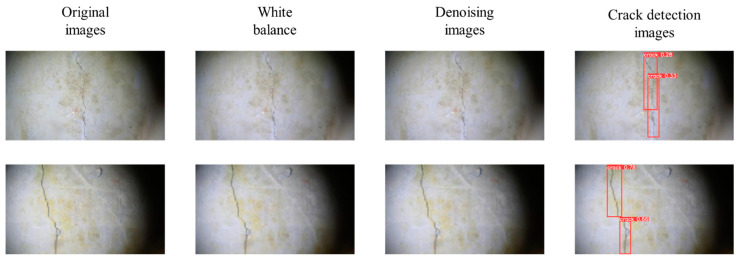
Crack detection results of two experimental images.

**Figure 25 sensors-24-05981-f025:**
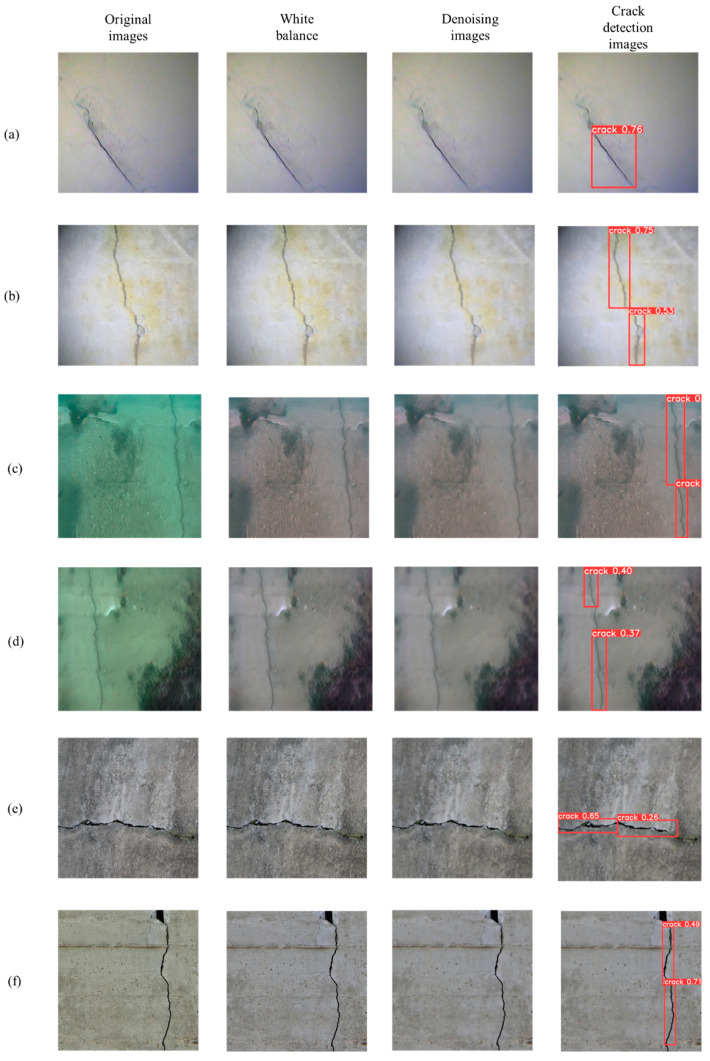
The detection process of other underwater and surface crack images: (**a**,**b**) underwater crack images under low light conditions; (**c**,**d**) underwater crack images with significant color differences; (**e**,**f**) above-water images.

**Table 1 sensors-24-05981-t001:** Image metrics corresponding to different white balance methods (Figure 16a).

Processing Method	UICM	UISM	UIConM	UIQM	PSNR	SSIM
Original	14.897	0.136	−0.119	0.032	-	-
Mean white balance	17.441	**0.145**	−0.118	0.111	30.591	0.987
Perfect reflection	18.119	0.140	−0.126	0.102	30.102	0.985
Ours	**18.395**	0.140	−0.118	**0.136**	**39.783**	**0.988**
Color cast	17.260	0.133	**−0.112**	0.123	37.406	0.987
Gray world	17.551	0.141	−0.122	0.099	22.425	0.977

**Table 2 sensors-24-05981-t002:** Image metrics corresponding to different white balance methods.

Original Image	Processing Method	UICM	UISM	UIConM	UIQM	PSNR	SSIM
Figure 16a	Original	14.897	0.136	−0.119	0.032	-	-
	Mean white balance	17.441	**0.145**	−0.118	0.111	30.591	0.987
	Perfect reflection	18.119	0.140	−0.126	0.102	30.102	0.985
	**Ours**	**18.395**	0.140	−0.118	**0.136**	**39.783**	**0.988**
	Color cast	17.260	0.133	**−0.112**	0.123	37.406	0.987
	Gray world	17.551	0.141	−0.122	0.099	22.425	0.977
Figure 16b	Original	15.206	0.199	−0.154	−0.063	-	-
	Mean white balance	17.314	**0.213**	−0.144	0.035	21.771	0.980
	Perfect reflection	16.167	0.201	−0.150	−0.017	16.903	0.948
	**Ours**	**17.995**	0.200	**−0.144**	**0.051**	**27.960**	**0.982**
	Color cast	17.656	0.198	−0.144	0.041	27.237	**0.982**
	Gray world	15.71	0.156	−0.146	−0.021	11.278	0.890
Figure 16c	Original	18.227	0.260	−0.142	0.082	-	-
	Mean white balance	17.131	**0.277**	−0.143	0.054	25.738	0.983
	Perfect reflection	17.706	0.266	−0.149	0.045	24.052	0.975
	**Ours**	**18.412**	0.252	−0.143	**0.084**	**38.176**	**0.984**
	Color cast	17.662	0.256	**−0.139**	0.075	37.244	**0.984**
	Gray world	17.642	0.263	−0.144	0.060	19.532	0.964
Figure 16d	Original	17.557	0.126	−0.122	0.094	-	-
	Mean white balance	17.341	**0.138**	−0.123	0.089	28.469	0.990
	Perfect reflection	17.457	0.130	−0.134	0.051	29.082	0.988
	**Ours**	**18.230**	0.111	−0.123	**0.112**	**39.023**	**0.991**
	Color cast	17.982	0.123	**−0.120**	0.105	36.371	0.990
	Gray world	18.034	0.129	−0.125	0.106	31.482	0.989
Figure 16e	Original	16.850	0.158	−0.127	0.069	-	-
	Mean white balance	17.425	**0.163**	−0.125	0.093	28.731	**0.984**
	Perfect reflection	17.262	0.161	−0.138	0.099	23.159	0.974
	**Ours**	**18.037**	0.159	−0.124	**0.112**	**38.929**	**0.984**
	Color cast	16.730	0.156	**−0.121**	0.087	37.449	**0.984**
	Gray world	16.187	0.157	−0.125	0.054	16.130	0.949

**Table 3 sensors-24-05981-t003:** **1.** Image metrics of different image denoising methods in Figure 19a–c. **2.** Image metrics of different image denoising methods in Figure 19d,e.

1							
Input Image	Processing Method	UICM	UISM	UIConM	UIQM	PSNR	SSIM
Figure 19a	Input	18.395	0.140	−0.118	0.136	39.783	**0.988**
	**Bilateral filtering**	18.333	0.124	**−0.114**	**0.145**	**40.326**	**0.988**
	Median filtering	18.399	0.134	−0.118	0.138	39.768	0.975
	Mean filtering (1 × 1)	**18.404**	0.140	−0.119	0.136	39.778	**0.988**
	Mean filtering (3 × 3)	18.388	**0.141**	−0.118	0.137	39.723	0.987
	Mean filtering (5 × 5)	18.347	0.135	−0.116	0.141	39.265	0.987
	Mean filtering (7 × 7)	18.329	0.122	−0.114	0.144	38.539	0.972
	Gaussian filtering (1 × 1)	18.398	0.140	−0.119	0.136	39.778	**0.988**
	Gaussian filtering (3 × 3)	18.395	0.139	−0.118	0.137	39.799	**0.988**
	Gaussian filtering (5 × 5)	18.386	0.135	−0.117	0.138	39.674	0.986
	Gaussian filtering (7 × 7)	18.365	0.130	−0.116	0.140	39.413	0.983
Figure 19b	Input	17.995	**0.200**	−0.144	0.051	27.960	0.982
	**Bilateral filtering**	17.991	0.129	**−0.137**	**0.055**	**27.990**	**0.982**
	Median filtering	17.988	0.174	−0.142	0.051	27.921	0.942
	Mean filtering (1 × 1)	**17.993**	**0.200**	−0.144	0.052	27.960	0.974
	Mean filtering (3 × 3)	17.990	0.174	−0.142	0.051	27.914	0.971
	Mean filtering (5 × 5)	17.990	0.148	−0.139	0.052	27.832	0.971
	Mean filtering (7 × 7)	17.990	0.128	−0.137	**0.055**	27.750	0.935
	Gaussian filtering (1 × 1)	**17.993**	**0.200**	−0.144	0.052	27.960	0.982
	Gaussian filtering (3 × 3)	17.985	0.177	−0.142	0.051	27.941	0.977
	Gaussian filtering (5 × 5)	17.989	0.165	−0.141	0.051	27.911	0.970
	Gaussian filtering (7 × 7)	17.986	0.149	−0.140	0.052	27.868	0.960
	Input	18.412	**0.252**	−0.143	0.084	38.176	**0.984**
Figure 19c	**Bilateral filtering**	18.387	0.185	**−0.135**	**0.091**	**38.203**	**0.984**
	Median filtering	18.405	0.230	−0.141	0.084	38.070	0.946
	Mean filtering (1 × 1)	**18.419**	**0.252**	−0.143	0.084	38.167	0.982
	Mean filtering (3 × 3)	18.386	0.234	−0.141	0.084	37.984	0.981
	Mean filtering (5 × 5)	18.391	0.209	−0.138	0.088	37.018	0.981
	Mean filtering (7 × 7)	18.381	0.181	**−0.135**	0.090	35.807	0.936
	Gaussian filtering (1 × 1)	**18.419**	0.252	−0.143	0.084	38.167	**0.984**
	Gaussian filtering (3 × 3)	18.401	0.235	−0.141	0.084	38.123	0.983
	Gaussian filtering (5 × 5)	18.394	0.223	−0.140	0.085	37.840	0.978
	Gaussian filtering (7 × 7)	18.393	0.205	−0.138	0.086	37.285	0.968
**2**							
**Input Image**	**Processing Method**	**UICM**	**UISM**	**UIConM**	**UIQM**	**PSNR**	**SSIM**
Figure 19d	Input	**18.23**	0.111	−0.123	0.112	39.023	**0.991**
	**Bilateral filtering**	18.018	0.131	**−0.122**	**0.114**	**39.054**	**0.991**
	Median filtering	18.023	0.113	−0.123	0.102	39.048	0.983
	Mean filtering (1 × 1)	18.04	0.111	−0.123	0.102	39.013	0.991
	Mean filtering (3 × 3)	18.01	0.129	−0.124	0.103	39.046	0.99
	Mean filtering (5 × 5)	18.019	**0.139**	−0.123	0.109	38.796	0.99
	Mean filtering (7 × 7)	18.021	0.127	−0.122	0.11	38.335	0.982
	Gaussian filtering (1 × 1)	18.04	0.111	−0.123	0.102	39.013	0.991
	Gaussian filtering (3 × 3)	18.028	0.12	−0.124	0.102	39.077	0.991
	Gaussian filtering (5 × 5)	18.025	0.124	−0.123	0.104	39.025	0.99
	Gaussian filtering (7 × 7)	18.027	0.123	−0.123	0.106	38.883	0.988
Figure 19e	Input	18.037	0.159	−0.124	0.112	38.929	**0.984**
	**Bilateral filtering**	18.003	0.113	**−0.117**	**0.124**	**38.943**	**0.984**
	Median filtering	18.033	0.143	−0.122	0.115	38.697	0.952
	Mean filtering (1 × 1)	**18.043**	0.16	−0.124	0.112	38.928	**0.984**
	Mean filtering (3 × 3)	18.015	0.147	−0.123	0.113	38.645	0.979
	Mean filtering (5 × 5)	17.995	0.128	−0.119	0.118	37.817	0.979
	Mean filtering (7 × 7)	17.994	0.113	**−0.117**	**0.124**	37.062	0.946
	Gaussian filtering (1 × 1)	**18.043**	**0.16**	−0.124	0.112	38.928	**0.984**
	Gaussian filtering (3 × 3)	18.026	0.147	−0.123	0.113	38.827	0.982
	Gaussian filtering (5 × 5)	18.02	0.139	−0.121	0.115	38.539	0.977
	Gaussian filtering (7 × 7)	17.996	0.128	−0.119	0.118	38.098	0.969

**Table 4 sensors-24-05981-t004:** Image metrics corresponding to different image denoising methods.

Original Image	Processing Method	UICM	UISM	UIConM	UIQM	PSNR	SSIM
Figure 16a	Original	14.897	0.136	−0.119	0.032	-	-
	Processed images	18.333	0.124	−0.114	0.145	40.326	0.988
Figure 16b	Original	15.206	0.199	−0.154	−0.063	-	-
	Processed images	17.991	0.129	−0.137	0.055	27.990	0.982
Figure 16c	Original	18.227	0.260	−0.142	0.082	-	-
	Processed images	18.387	0.185	−0.135	0.091	38.203	0.984
Figure 16d	Original	17.557	0.126	−0.122	0.094	-	-
	Processed images	18.018	0.131	−0.122	0.114	39.054	0.991
Figure 16e	Original	16.850	0.158	−0.127	0.069	-	-
	Processed images	18.003	0.113	−0.117	0.124	38.943	0.984

**Table 5 sensors-24-05981-t005:** Training parameters.

Parameter Name	Meaning	The Set Value
Hardware configuration	Computer parameters	12th Gen Intel(R) Core(TM) i7-12700F 2.10 GHz (Intel, Santa Clara, CA, USA)
Environment configuration	Virtual environment	Python 3.8, PyTorch==1.13, CUDA 11.7
Batch size	Input size	4
Workers	Number of data-loading threads	4
Epochs	Iterations	200
lr0	Initial learning rate	0.01
lrf	Cosine annealing hyperparameter	0.01
Momentum	Learning rate momentum	0.937
weight_decay	Weight attenuation coefficient	0.0005

**Table 6 sensors-24-05981-t006:** Comprehensive test results of different training models.

Model Name	*P*	*R*	*F* _1_	*mAP* _0.5_	*mAP* _0.5:0.95_
YOLOv9	73.2%	68.1%	0.710	74.1%	42.2%
YOLOv9-e	70.2%	68.0%	0.690	73.0%	41.3%
YOLOv9-c	66.0%	68.9%	0.670	69.4%	38.4%
YOLOv9-m	**74.1%**	65.9%	0.700	72.2%	41.7%
YOLOv9-s	68.6%	69.5%	0.680	72.7%	41.5%
YOLOv9-t	64.1%	66.7%	0.650	69.7%	38.3%
YOLOv9-OREPA	72.4%	**73.6%**	**0.730**	**74.8%**	**44.1%**

## Data Availability

Data are contained within the article.

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
