# Peer review of "An Underwater Crack Detection System Combining New Underwater Image-Processing Technology and an Improved YOLOv9 Network"

_sensors, 2024, doi:10.3390/s24185981_

Round 1

Reviewer 1 Report

Comments and Suggestions for Authors

The refined comment:
This manuscript titled "An underwater crack detection system combining new underwater image processing technology and improved YOLOv9 network" addresses a new type of underwater crack detection method is provided, and a novel detection approach is proposed to convert underwater images into underwater images through image processing technology, avoiding the problem of insufficient underwater crack datasets.

It is an innovative and practical underwater crack detection method. Overall, this manuscript has made significant progress in the field of using computer vision for structural health monitoring. With suggested revisions and additional details, this paper will be highly suitable for publication in Sensors.

1. The comparison of network improvements is not obvious: Figure 23 does not illustrate the advantages of YOLOv9 OREPA over other YOLO series in actual detection, such as missing or missed detections.

This section should be added to illustrate the advantages.
2. Spelling errors and clarity issues in the graphics: In Figure 23, it should be YOLOv9 OREPA instead of YOLOv9 OPERA The resolution of Figure 21. (a) is low and should be replaced with a high-resolution image.

3. Font issue in formulas: TP and FP in formulas 26 and 27 should be italicized.

4. Ensure that every font has meaning: YOLO on line 114 and OREPA on line 345 are both new to the article and should be given specific meanings for more people to understand.

5. Some writing errors: Spaces should be added before the font and parentheses in Table 3.

6. Please add some limitations and future work.

Reviewer 2 Report

Comments and Suggestions for Authors

the manuscript proposed an image processing method that transfrom underwater crack images into above water images with original creak features. Then detected the crack based on improved YOLOV9 Model. The method proposed here are robust and the results are reasonable. However, i have some concerns.

1, There are too many words in this presented paper, which does not match the importance of this paper. In this paper, the basic concepts and formulas are numerous, please delete them and reorganize the whole structure.

2,some references are highly related to your paper, you can learn and cite them. 

1. Courtial, A., Touya, G. & Zhang, X. Constraint-Based Evaluation of Map Images Generalized by Deep Learning. Journal of Geovisualization & Spatial Analysis, 2022,6(1):13. https://doi.org/10.1007/s41651-022-00104-2

Reviewer 3 Report

Comments and Suggestions for Authors

This paper presents an innovative approach to underwater crack detection by combining advanced image processing techniques with an improved YOLOv9 network, specifically tailored to the underwater environment. The system is designed to overcome the challenges of low-quality underwater images, such as noise. Experimental results are presented, showcasing improved performance in terms of image quality metrics and detection accuracy over existing methods. Here are several suggestions to improve this manuscript.

1. In the introduction (abs abstract), the strength of OREPA is not mentioned, which however should be introduced as there should be motivation of using it.

2. The regression loss of eq 18 should be better explained, as it is now very confusing and does not have a clear definition.

3. Please check the reference at line 275.

4. Could the authors justify why the batch size is chosen as 4, which seems to be a very small number? This applies to other settings, for instance, momentum of 0.937, which seems to be a very specific value.

5. In Table 2, the mean white balance method always outperforms the proposed method under UISM (as well as color case under UIQM). It would be better if the authors can elaborate on this. 

6. Does the dataset include real-world images from different underwater environments (e.g., oceans, lakes, dams), or is it limited to controlled laboratory conditions? How does the method perform on larger, more diverse datasets?

7. What are the computational resource requirements (in terms of memory, processing power, and time) for the YOLOv9-OREPA model and other YOLOv9 models? Since the authors argue YOLOv9-OREPA is more efficient in terms of momery, it should be presented.

Reviewer 4 Report

Comments and Suggestions for Authors

The review can be found in the attachment

Comments on the Quality of English Language

The quality of the English is quite good

Round 2

Reviewer 2 Report

Comments and Suggestions for Authors

Underwater crack detection is challenging and meaningful. This study combines a white balance method and a bilateral filtering denoising method to solve problems. Overall. the paper is well-structured and readable. However, there are some comments and suggestions for authors.

1. There are differences between the underwater and above-water environments, and are your underwater data processing method solves these problems?

2. L110. After summarizing the problems, the algorithm of this paper should be pointed out, and the advantages of the algorithm need to be explained.

3. Section 4.1. “After capturing multiple images, five images in Fig. 16 with obvious crack features were selected for use in the experiment.” Why did you choose 5 images? Are these images representative?

4. One reference is recommended to help increase the added value of the introduction and discussion.

Zhao, Y., Wang, G., Yang, J. et al. AU3-GAN: A Method for Extracting Roads from Historical Maps Based on an Attention Generative Adversarial Network. Journal of Geovisualization & Spatial Analysis, 2024,8(1):26. https://doi.org/10.1007/s41651-024-00187-z.

Comments on the Quality of English Language

Minor English modifications are required.
